# Outcome of Recipient Surgery and 6-Month Follow-Up of the Swedish Live Donor Robotic Uterus Transplantation Trial

**DOI:** 10.3390/jcm9082338

**Published:** 2020-07-22

**Authors:** Mats Brännström, Pernilla Dahm-Kähler, Jana Ekberg, Randa Akouri, Klaus Groth, Anders Enskog, Verena Broecker, Johan Mölne, Jean-Marc Ayoubi, Niclas Kvarnström

**Affiliations:** 1Department of Obstetrics & Gynecology, Sahlgrenska Academy, University of Gothenburg, SE-41345 Göteborg, Sweden; pernilla.dahm-kahler@vgregion.se (P.D.-K.); randa.akouri@vgregion.se (R.A.); klaus.groth@vgregion.se (K.G.); 2Department of Transplantation, Sahlgrenska Academy, University of Gothenburg, SE-41345 Göteborg, Sweden; jana.ekberg@vgregion.se (J.E.); niclas.kvarnstrom@vgregion.se (N.K.); 3Department of Anesthesiology and Intensive Care, Sahlgrenska Academy, University of Gothenburg, SE-41345 Göteborg, Sweden; anders.enskog@vgregion.se; 4Department of Laboratory Medicine, Sahlgrenska Academy, University of Gothenburg, SE-41345 Göteborg, Sweden; verena.brocker@vgregion.se (V.B.); johan.molne@vgregion.se (J.M.); 5Department of Obstetrics, Gynecology and Reproductive Medicine, Hospital Foch, Suresnes and University of Versailles, 92150 San-Quentin en Yvelines, France; jm.ayoubi@hopital-foch.org

**Keywords:** human, infertility, recipient, transplantation, uterus

## Abstract

Uterus transplantation has proved to be a feasible treatment for uterine factor infertility. Herein, we report on recipient outcome in the robotic uterus transplantation trial of 2017–2019. The eight recipients had congenital uterine aplasia. The donors were six mothers, one sister, and one family friend. Donor surgery was by robotic-assisted laparoscopy. Recipient surgery was by laparotomy and vascular anastomoses to the external iliacs. The duration (median (ranges)) of recipient surgery, blood loss, measured (left/right) uterine artery blood flow after reperfusion, and length of hospital stay were 5.15 h (4.5–6.6), 300 mL (150–600), 43.5 mL/min (20–125)/37.5 mL/min (10–98), and 6 days (5–9), respectively. Postoperative uterine perfusion evaluated by color Doppler showed open anastomoses but restricted blood distribution in two cases. Repeated cervical biopsies in these two cases initially showed ischemia and, later, necrosis. Endometrial growth was not seen, and hysterectomy was later performed, with pathology showing partly viable myometrium and fibrosis but necrosis towards the cavity. The other six patients acquired regular menstrual cyclicity. Surgery was performed in two patients to correct vaginal stenosis. Reversible rejection episodes were seen in two patients. In conclusion, the rate of viable uterine grafts during the initial 6-months of the present study (75%) leaves room for improvement in the inclusion/exclusion criteria of donors and in surgical techniques. Initial low blood flow may indicate subsequent graft failure.

## 1. Introduction

Uterus transplantation (UTx), as a treatment for absolute uterine factor infertility (AUFI), proved its feasibility by the first live birth after UTx in 2014 [1], which involved the fifth UTx procedure within the first clinical trial of UTx, with surgeries performed in Sweden in 2012–2013 [2]. Several more births have followed, after both live-donor UTx [3,4] and deceased-donor procedures [5,6]. 

All initial live-donor UTx trials were with the laparotomy technique of the donor. The main difficulty of the laparotomy technique in donor hysterectomy is the dissection of the distal ureters and the deep uterine veins. These anatomical structures are firmly attached to the para-vaginal/cervical tissue, and the venous outflow from the deep uterine veins often includes an extensive network of venous branches riding over and under the ureter. It has been proposed that minimally invasive surgery, especially by robotic-assisted laparoscopy, may aid in these difficult and fine dissection procedures deep in the pelvis [7,8]. In 2015, the first case of a robotic-assisted donor hysterectomy was performed [9]. Donor surgery was fully robotic and with the retrieval of the uterus through the vagina. Due to difficulties in the dissection of the deep uterine veins, the venous outflow sections procured on the graft were solely the complete utero-ovarian veins on both sides. This required a bilateral oophorectomy in the premenopausal donor, which is a contentious procedure due to the increased risk of long-term morbidity and mortality that is secondary to the sudden cessation of hormonal output from the ovaries [10]. Our approach is to develop live donor robotic-assisted hysterectomy, with the procurement of the deep uterine veins and the proximal parts of the utero-ovarian veins [7]. This restricted recovery of only parts of the upper uterine venous outflow sections is compatible with preserved ovarian function. The purpose of procurement of both upper and lower uterine outflows (deep uterine veins) is to have several venous outflow options for the evaluation of the outflow and quality of veins on the back-table in order to select the most suitable veins for venous anastomoses.

Along these lines, we performed eight robotic-assisted donor hysterectomies in 2017–2019 and found an obvious evolution of the surgical procedures within the trial [11]. Thus, in the last three procedures, major parts of donor surgery, including full dissection of deep uterine veins, could be performed by robotically assisted surgery. Details on the donor surgery of these eight procedures, with emphasis on robotic surgery, have been extensively described in our recent article [11]. The present study describes the recipients, preoperative investigations, recipient surgery, and 6-month outcomes of the grafts of the Swedish robotic trial. To our knowledge, this is the largest trial of the use of minimally invasive surgery in UTx.

## 2. Material and Methods

### 2.1. Approval and Setting 

The study was approved by the Regional Ethics Committee (no. 362-16; 11 June 2016) and registered (NCT02987023). The setting was Sahlgrenska University Hospital, a tertiary hospital with a large transplant program and a specialized gyne-oncology division. The collaborative team has extensive experience in robotic surgery and UTx (>15 procedures prior to this trial). 

### 2.2. Patients and Preoperative Investigations 

The eight recipients had uterine agenesis as part of the Mayer-Rokitansky-Kuster-Hauser syndrome (MRKHs). The patients underwent meticulous medical and psychological investigations, including imaging by abdominal/pelvic magnetic resonance imaging (MRI), serology (hepatitis A/B, human immunodeficiency virus, cytomegalovirus, Epstein-Barr virus, syphilis), clinical chemistry, human leukocyte antigen (HLA) typing with assays of donor-specific antibodies (DSAs), psychological tests, and interviews. Gynecological examination of recipients included an evaluation of the vagina and tests for chlamydia, gonorrhea, and high-risk human papillomavirus. In vitro fertilization (IVF) was performed to cryopreserve at least eight embryos. This necessitated 1–3 IVF cycles, using a protocol with a random and simultaneous start of purified human menopausal gonadotropin (hMG; Menopur^®^; Ferring, Sweden) and 0.25 mg of the gonadotropin-releasing hormone (GnRH) antagonist ganirelix (Orgalutran^®^; Merck, Germany). 

### 2.3. Surgery

Donor surgery was partly performed by a robotic-assisted approach that has been described in detail elsewhere [11]. One important aspect of donor surgery that has not been specified previously is the technique for the division of the donor vagina, which may have consequences for the development of vaginal stenosis (see results in Section 3). Division of the vagina will, in general, give a vaginal cuff of 1–2 cm. In procedure #1–2 of the present study, the vagina, including the fascia and mucosa, was divided completely by the electrothermal bipolar Ligasure^®^ (Medtronic, Minneapolis, MN, USA) tissue-sealing system. In procedure #3–8, the vaginal fascia and mucosa were divided by scissors and with uni- and-bipolar diathermy for hemostasis since we observed significant vaginal stenosis in the initial two cases (see results in Section 3), possibly related to the use of Ligasure^®^.

Recipient surgery was started a short time before vascular clamping in the donor, and this provided sufficient time for surgical preparations of the recipient during back-table preparations. Back-table preparation was initiated by submerging the uterus in ice-slush and bilateral arterial flushing with 10 mL of physiological saline containing heparin (1250 IE) and lidocaine (50 mg), followed by perfusion with a minimum of 200 mL of cold histidine-tryptophan-ketoglutarate preservation solution. During the back-table perfusion, the four veins on the graft were assessed for outflow and quality in order to determine the suitability for anastomosis. Leakages from vessels were closed by 7-0 polypropylene sutures, and discarded outflow options were closed. The adjacent segment of the internal iliac vein was trimmed, and minimal dissection of the proximal part of the utero-ovarian vein was performed. 

The surgery of the recipient was through a pubo-umbilical midline incision. The preparatory surgery of the recipient is illustrated in Figure 1A. Initially, the vaginal vault is exposed by midline cleavage of the rudimentary uterus and dissection of the posterior aspect of the bladder fundus, which, in general, is covering the top of the neovagina. The rectum is then dissected from the posterior aspect of the vagina. The dissection to expose the vault of the neovagina, which, in general, is shorter than a normal vagina, is aided by the use of a spherical vault-presentation silicon probe (30103^®^; Karl Storz, Germany), which is placed inside the vagina and pushed upwards in the direction of the umbilicus. When the vaginal vault is freed from the bladder and the rectum by a sagittal distance of around 5 cm, the dissection is sufficient to permit the subsequent opening and vaginal-vaginal anastomosis. The surgery is then directed towards the dissection of the vessels for the preparation of vascular anastomosis sites. The external iliac arteries and veins are dissected bilaterally, and then fixation sutures (1-0; nonresorbable), to be used for subsequent uterine fixation, are placed bilaterally into the sacrouterine ligaments, round ligaments, and the bisected uterine rudiment (Figure 1A). 

The chilled graft is then placed into its position inside the pelvis, and the vascular anastomoses are completed between the segments of the internal iliac vessels of the graft and the proximal portions of the utero-ovarian veins and external vessels of the recipient by use of continuous 6-0 and 7-0 polypropylene sutures for arteries and veins, respectively (Figure 1B). In some procedures, the proximal portion of the utero-ovarian vein is coupled to the internal iliac vein segment on the back-table (Figure 1B). All anastomoses are completed before revascularization. After proper uterine perfusion has been confirmed by Doppler measurement (Vascular TIFM Probe^®^; Medistim, Norway), the vagina of the recipient is opened by a sagittal incision. End-to-end anastomosis is then performed between the vaginal vault of the recipient and the vaginal rim of the graft by a continuous 2-0 resorbable suture. Fixation sutures, placed in the recipient as described above, are then sutured to the respective parts of the graft, and a large bladder peritoneal flap of the graft is sutured on top of the bladder. Prophylaxis against infections and thrombosis was with piperacillin/tazobactam 4 g i.v. (intravenously) every 8 h for 4 days and 5000 IU dalteparin s.c. (subcutaneously) for 6 weeks, plus 75 mg of acetylsalicylic acid until graft removal, respectively. 

### 2.4. Immunosuppression and Post Transplantation Follow-Up 

The immunosuppression was an induction protocol with 500 mg i.v. methylprednisolone (Solu-Medrol^®^; Pfizer, USA) and 20 mg basiliximab (Simulect^®^; Novartis, Switzerland) just prior to reperfusion, with a second dose of 20 mg basiliximab on day 4. Oral prednisolone (Prednisolone^®^; Pfizer) was used at 80 mg on day 2 and then tapered to withdrawal on day 6. Maintenance immunosuppression was by oral administration of tacrolimus (Adport^®^; Sandoz, Switzerland) twice daily from the day of surgery. Target trough levels of tacrolimus were 10, 8, and 5–7 ng/mL, during months 1, 2–3, and 4–6, respectively. Additional maintenance immunosuppression was azathioprine (2 mg/kg/day; Imurel^®^; Orion Pharma, Finland).

### 2.5. Post-Transplantation Follow-Up 

The graft was evaluated daily by assessment of blood flow (Doppler by abdominal probe) during the hospital stay and by gynecological examination (including biopsy) at day 4–5 after UTx. The patients were then, during the initial postoperative month, seen weekly for gynecological examination, including inspection of the cervix, vaginal bacterial culture, and cervical biopsy for rejection diagnosis. Grading of rejection was performed according to our uterine rejection grading system [12]. These gynecological examinations, with cervical biopsies, were after the initial postoperative month performed every 4 weeks during the study period. If rejection or any other pathology was seen, a follow-up biopsy was taken two weeks after rejection treatment was initiated, and if the rejection had reversed, the increase in immunosuppression was gradually withdrawn. If the second biopsy showed rejection, immunosuppression was continued or further increased, and another biopsy was taken two weeks later. 

## 3. Results

### 3.1. Preoperative Findings

Data concerning characteristics of recipients and donors, with the results of their preoperative investigations, are given in Table 1. The eight recipients (age 22–33 years at transplantation) had uterine agenesis as part of the Mayer-Rokitansky-Küster-Hauser syndrome (MRKHs), and all had bilateral kidneys (MRKHs type 1). Seven of the recipients had dilated vaginas, and one had undergone surgery by McIndoe split-skin graft vagina. The eight donors were six mothers (45–62 years old), one sister (37 years old), and one family friend (48 years old), with the three donors above 50 years of age (all mothers) being postmenopausal. The two donating mothers of age 55 and 57 years had their last menstrual periods around 1.5 years before transplantation. They were given sequential menopausal hormonal therapy from three months before donor hysterectomy. The oldest donor (62 years at uterus donation) had been on sequential menopausal hormonal therapy since she acquired perimenopausal symptoms at 50 years of age. The donors had delivered 1–4 children. All were vaginal deliveries, with normal birth weights (Table 1), and at term. All donors except donor #5 were never-smokers, and donor #5 was only an occasional smoker, with no smoking at all during three months preceding surgery. All donors and recipients were serologically positive for Epstein-Barr virus (EBV), and the status of cytomegalovirus (CMV) antibodies varied (Table 1). Data concerning blood groups and HLA matching are also given in Table 1. No recipient had DSA at transplantation.

### 3.2. Surgical Data

The data on the surgeries are summarized in Table 2. The total surgical time, from first skin incision to last skin suture, varied between 4.5 and 6.6 h (median 5.15 h). The total ischemia time, defined as from the time of the cross-clamping of vessels in the donor to the reperfusion of the uterus was established in the recipient, varied from 1.7 to 3.2 h (median 2.1 h). There was a wide variation in immediate perioperative reperfusion blood flow, from 125/98 mL/min in patient #6 to 20/10 mL/min in patient #3. 

Patient #3 had the longest surgical time (6.6 h) and the longest total graft ischemia time (3.2 h) before completed vascularisation. In this case, the arteries thrombosed spontaneously during the uterine fixation and completion of the vaginal anastomosis. This was discovered during the second routine flow measurement before the closure of the abdomen, with the graft fixed in its position. Both arterial anastomoses had to be opened, and thrombus formations were carefully removed, followed by extensive flushing with heparin solution. The veins remained open, and after the flushing, there was a backflow in the arteries. After completed reanastomosis of both arteries, blood flow in both uterine arteries was confirmed with flow-measurement. The time of warm ischemia during the thrombectomy was around 1.5 h.

Two arterial anastomoses (Figure 1B) were used in every surgery (Table 2). Three outflow veins were used in four cases, and four veins were used in the other four cases. The options for vein anastomoses, also using the proximal part of the utero-ovarian vein (see Materials and Methods), are illustrated in Figure 1B. The estimated blood loss (EBL) during surgery varied between 100 and 600 mL (median 150 mL). The length of hospital stay (LOS) was between 5 and 9 days.

During the hospital stay, the daily transabdominal color Doppler ultrasound indicated a diminished distribution of blood flow in the central portions of the uteri of two patients (#3, # 8) and initial biopsies around day 4–5 showed focal ischemia in the ectocervix of these patients. 

### 3.3. Post-Transplantation Period

Data concerning post-transplantation follow-up are given in Table 3. Six out of the eight patients had regular menstruation, starting three to six weeks after UTx. In four out of these six patients with regular menstruation, no rejection episodes were seen during the observation period of 6 months. 

Two patients had one rejection episode each (Table 3). The graft of patient #1 showed a Grade 1 rejection three months after UTx (Figure 2A), and a Grade 2 rejection was seen in patient #4 four months after UTx (Table 3). Analysis of DSA was not performed on these occasions since there were no indications of humoral rejection. Treatments with oral corticosteroids were given until a follow-up biopsy was normal. The levels of hemoglobin and creatinine, and trough levels of tacrolimus were relatively stable (Table 3). 

Patients #1 and #2 showed gradual stenosis over the vaginal-vaginal anastomosis line. When the diameter at the stenosis was less than 10 mm, vaginal surgery by diathermy was performed (2 and 5 months postoperatively in patients #1 and #2, respectively) to open the stenosis to allow further inspection of the cervix and for biopsies to be taken. No other patients developed vaginal stenosis.

The color Doppler ultrasound could only detect blood flow in the major uterine vessels in patients #3 and #8, and the uterine size decreased gradually. Repeated cervical biopsies, starting from the first postoperative biopsy, showed ischemia followed by necrosis, and core biopsies of the myometrium showed patchy necrosis. Hysteroscopy with biopsy was performed around 6 months post-UTx in patient #3, which did not show any viable endometrium. Prior treatment with a high dose (4 mg daily) of estradiol orally for 3 weeks had not shown any visible endometrial lining in repeated transvaginal ultrasound examinations. Hysterectomy was performed soon afterwards. The explanted uterus had a length of 5 cm and was anatomically distorted. Microscopically, no endometrium was found. The myometrium was atrophic, and there were focal infarcts with granulation tissue. Larger arteries were occluded by sclerotic intimal tissue with some foam cells but without active endarteritis.

In the uterine graft of patient #8, there was also the development of decreased blood flow, with core biopsies showing ischemic changes in the myometrium and ectocervical biopsies, initially exhibiting ischemia, followed by patchy necrosis. Hysteroscopy at one month after UTx could not detect any endometrium. Based on the experience of patient #3 (see above), hysterectomy was performed shortly after hysteroscopy. The explanted uterus measured 8.5 cm in length. On sectioning, the inner half of the uterus wall and the cervix appeared largely necrotic (Figure 2B). Microscopically (Figure 2B), the endometrium and inner half of the myometrium were necrotic, the outer half showed partly viable myometrium. Arteries showed moderate intimal sclerosis, with some atheromatous plaques but no endarteritis. Several veins showed partly organized thrombi. 

Both patients, with graft removal, had uneventful posthysterectomy periods. 

## 4. Discussion

Uterus transplantation is the first available treatment for AUFI, with previous motherhood options for this group being restricted to adoption or pregnancy in a gestational surrogate carrier. The proof-of-concept of UTx as a fertility treatment came with the first live birth after UTx, which occurred in 2014 [1]. Although the activity within the UTx field is high, there are only a limited number of published studies concerning short- and long-term follow-up in recipients after UTx. In the first UTx study performed in 2012–2013 in Sweden, the 6- and 12-month [13] outcomes of the nine recipients were reported in detail. Briefly, seven out of nine grafts were preserved, with regular menstrual patterns during the initial 1-year follow-up time [13]. Moreover, outcomes of parts of the study populations of recipients were reported by live donor UTx studies in the USA [14] and Germany [15], as well as from a mixed live- and deceased-donor UTx study in the Czech Republic [16]. The number of reported recipients in the UTx studies of the USA, Germany, and the Czech Republic were 5, 2, and 9, respectively. Thus, there are limited data available for comparative analysis of the results of the present study. 

Noteworthy is that UTx has some special features when compared to most other types of solid organ transplantations. Firstly, UTx is a quality-of-life enhancing transplantation rather than a vital type of transplantation (liver, heart, lung). Moreover, UTx is unique as an ephemeral transplantation, with the graft not being intended for life-long use, but rather, for a restricted time until one or two children have been delivered. This transient transplantation period will decrease the time the recipient is exposed to immunosuppression, with its associated side effects. Furthermore, after transplantation of the uterus, there is a long time-interval of at least a year from transplantation until proof of full functionality by a live birth. 

The major and clinically most important result of the present study is the rate of surgically successful grafts, with resumed menstrual cyclicity of the uterus being a notable sign of functionality. However, pregnancy and delivery of a healthy child is the final proof of a fully functional graft. Our accumulated experience in the UTx field tells us that a graft that shows viability during the initial 6 months will continue to be viable despite rejection episodes or other forms of tissue stress. However, most likely, not all viable grafts will result in a clinical pregnancy and live birth, as illustrated by the first surgically successful deceased donor UTx case from 2011, which presented with early miscarriages but no live birth reported yet [17]. The expected cumulative live birth rate in patients with surgically successful uterine grafts, who would be able to undergo multiple embryo transfers should be around 60%, similar to a normal IVF population that undergoes three IVF cycles with associated fresh and frozen embryo transfers [18].

In the present study, we experienced two rejection episodes during the initial six months in the six patients that had functional grafts during the entire follow-up period. This is similar to what we experienced in the initial Swedish trial of 2013, where three out of seven patients with functional grafts had reversible rejection episodes during the corresponding time period [12]. The histological analysis of the rejection biopsies did not show any signs of humoral rejection, and DSA was not analyzed. We did not perform any regular analyses of the development of DSA during the study period. 

The 6-month graft survival of the present study was 75%. In comparison, the graft survival rates from the original Swedish study [2] and the UTx studies of the USA [14], Germany [15], and the Czech Republic [16] were 78% (7/9), 40% (2/5), 100% (2/2), and 67% (6/9), respectively. Thus, a collective take-home-message from the results of these early studies, including the present study, is that improvements in the graft survival of UTx are needed. Developments in preoperative screening procedures to exclude suboptimal organs/patients and improvement of surgical and medical techniques will most likely be the base for further progress. Scientific studies to report outcomes, both negative and positive, are needed to increase the knowledge base, which is imperative to develop measures to increase the safety and efficiency of the UTx procedure. 

The two graft failures of the present study shared some features. The perioperative perfusion blood flows were lowest in the two patients with graft failures, exhibiting mean (right and left) values of 15 and 24.5 mL/min in patient #3 and #8, respectively. In the six successful grafts, the mean blood flow after reperfusion ranged from 39 to 112 mL/min. Moreover, it is of interest to keep in mind that the early graft failure (because of vascular thrombosis within the first week) of our original study [2] showed a maximum arterial flow after reperfusion of 10 mL/min, while successful grafts ranged between 30 and 75 mL/min. Blood flow measurements have not been published in the live donor UTx studies of the USA, Germany, and the Czech Republic [14,15,16]. Further studies have to determine whether low perfusion flow indicates a higher risk of graft failure. Moreover, there may be room for the development of more advanced assessments of perfusion capacity of the uterus on the back-table in order to only transplant organs with good perfusion after transplantation. In addition, it is of note that one additional recovered donor graft of the German live donor UTx study was not transplanted [15] because, during the back-table flushing, clear signs of impaired organ perfusion were already seen and a decision was taken to abort the transplantation. 

Even though the uteri of the two graft failures of the present study showed perfusion and blood flow postoperatively, the assessments with color Doppler during the initial weeks showed a clearly impaired distribution of blood to the central parts. Since the blood from the uterine arteries feeds blood to the subserosal part, and then continuation into the smaller spiral arteries ending in the endometrial stroma, this pattern of blood flow is expected at low perfusion. Another indication of insufficient blood flow in uterine areas far from the major branching of the uterine arteries is initial ischemia, followed by patchy necrosis and then confluent necrosis, as seen in repeated cervical biopsies. This observation is in line with the finding of early graft failure of the US study [15].

In the present study, we had a long lag phase from initial signs of irreversible hypoperfusion in case #3 until hysterectomy, but in the second graft failure (last case of the present study), the action to perform a hysterectomy was taken at a much earlier stage. This decision was established on the grounds of the morphology of early core biopsies, as well as hysteroscopy findings. Based on the posthysterectomy histopathology, it is very unlikely that the uterus would have shown functionality later.

The duration of recipient surgery in the present study ranged between 4.5 and 6.6 h, which should be compared to 4.2–5.9, 4.0–6.0, 4.5–6.0, 3.5–5, and 4.0–5.0 h in the original Swedish study [2], the US study [14], the German study [15], the Czech live-donor study-arm [16], and the Czech deceased-donor study-arm [16], respectively. Collectively, a surgical time on the recipient of 4–6 h seems appropriate, and differences in duration will naturally depend on anatomical variations, the extent of leakage from the graft at reperfusion, as well as the number of venous anastomoses to the external iliac veins. 

The total ischemic time, from the clamping of vessels in the donor until proper reperfusion in the recipient, varied between 1.7 and 3.2 h in the present study. The longest ischemic time (50 min more than the second-longest ischemia time of the present study) was in the patient with intraoperative arterial thrombosis, followed by thrombectomy. It is not likely that the recorded total ischemia time was the cause of the subsequent graft failure since the ischemic time of the second graft failure was much shorter (2 h). However, the long (1.5 h) warm ischemia, after thrombectomy in patient #3, may have had a negative impact on the outcome. It should be noted that the total ischemic time of the first successful deceased donor UTx procedure was as long as 7.8 h [5]. Ischemic times of 2.1–3.4, 4.0–5.6, 4.0–6.5, and 2.5–9.0 h were reported in the original Swedish study [2], the US study [14], the Czech live-donor study-arm [16], and the Czech deceased-donor study-arm [16], respectively.

One central observation from the present study is that vaginal stenosis was acquired in two patients, and stenosis was not experienced in any case of our original study [2]. Such stenosis prevents appropriate inspection of the uterine cervix and the attainment of a cervical biopsy for rejection diagnosis, and it also obstructs later embryo transfer procedures. Vaginal stenosis was also described in two out of five recipients in the live-donor arm of the Czech study [16]. We used an automatic electrothermal bipolar tissue-sealing system for transection of the vagina in the donor of the two cases with subsequent stenosis and then reverted to our original [2] method for division of the vagina. We speculate that the automatic electrothermal bipolar tissue-sealing system may have induced irreversible ischemic tissue damage, with fibrosis for a relatively large distance of the graft vagina, and that continuously activated fibroblasts in this area may have caused the stenosis. 

Another issue that has been a matter of debate in UTx is the venous outflow. In the present study, we used the deep uterine veins on segments of the internal iliac veins in every case, and in addition, one or two proximal parts of the utero-ovarian vein. A publication [4] of successful UTx, with live birth, using only two proximal parts of the utero-ovarian vein for venous anastomoses, exists. If future research demonstrates the feasibility of using only these veins as outflow, after a donor hysterectomy that would not need oophorectomy, this may greatly simplify uterus procurement from live donors. This fact is illustrated in the results of robotic-assisted uterus procurement of the donors with the recipients of the present study, where dissection of the uterine veins and their branches close to the distal ureter took 2–3 h [11], corresponding to 30–40% of the surgical time using the robot.

In the present study, two of the successful grafts were from postmenopausal donors. Sequential menopausal hormonal replacement therapy was used for both donors. For the oldest donor (62 years at uterus donation), hormonal therapy had been used for 7 years. It is likely that continuous estrogen therapy since the menopausal age of this donor is of importance for well-preserved uterine arteries. In the other postmenopausal donor (57 years old) with a successful graft, hormonal therapy was merely used to demonstrate functionality in terms of menstruation. It is unlikely that such a short course of hormone therapy would have had any beneficial effect on the uterine arteries, but rather, the relatively short hypo-estrogenic period did not have any major negative effects on the uterine vessels.

In conclusion, the present study of a complete study cohort of recipients for UTx shows that the surgical success rate needs to be improved. Low uterine blood flow at transplantation is an important early sign of subsequent graft failure. 

## Figures and Tables

**Figure 1 jcm-09-02338-f001:**
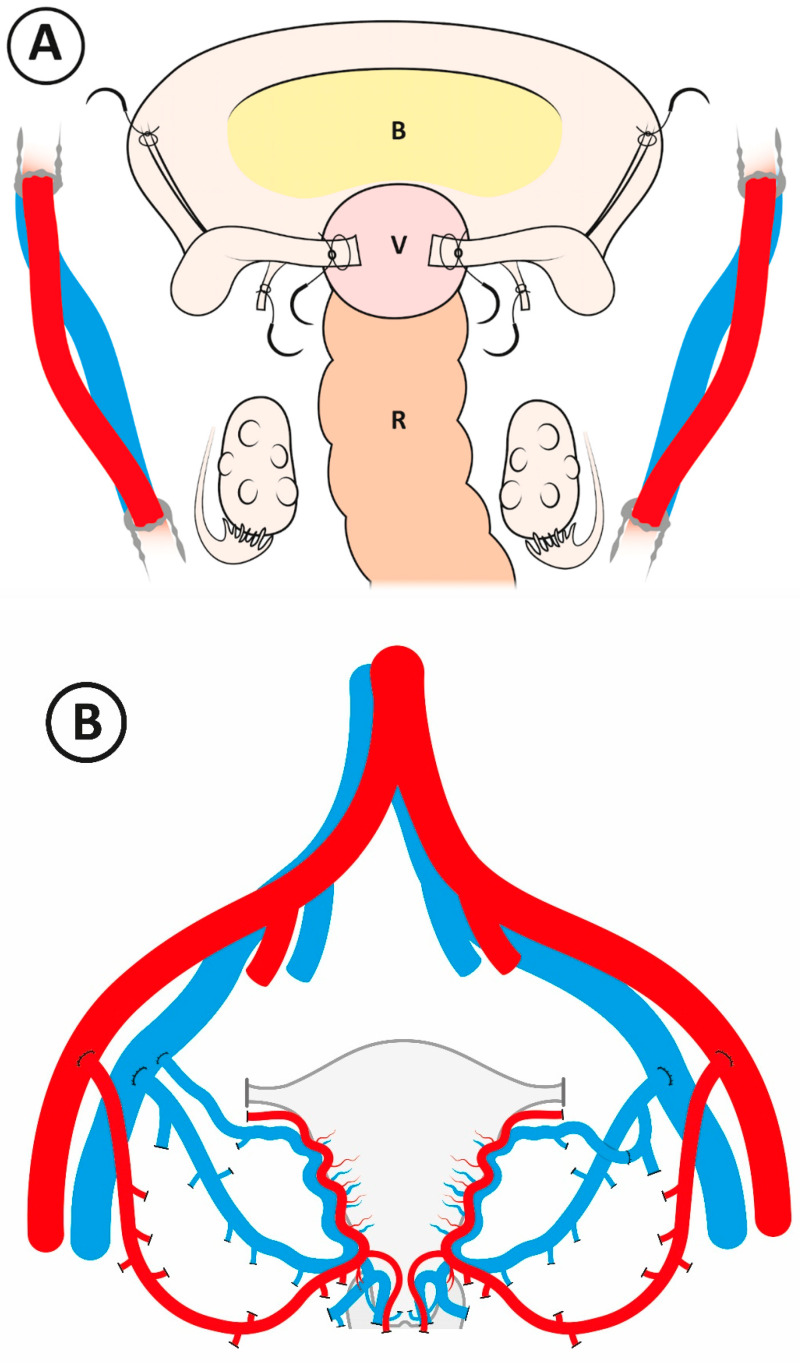
(**A**) Schematic graph illustrating surgical preparations in the recipient prior to the graft entering the pelvis. B = bladder, V = vaginal vault, R = rectum. (**B**) Principles for vascular anastomosis. The anterior portions of the internal iliac arteries are anastomosed end-to-side to the external iliac arteries. On the right side of the pelvis, a segment of the internal iliac vein, in continuation with the deep uterine vein, is anastomosed end-to-side to the external iliac vein. On the same side, the proximal part of the utero-ovarian vein is directly anastomosed end-to-side to the external iliac vein. On the left side of the pelvis, a segment of the internal iliac vein, in continuation with the deep uterine vein, is anastomosed end-to-side to the external iliac vein. On the same side, the proximal part of the utero-ovarian vein is anastomosed end-to-end to a branch of the donor internal iliac vein.

**Figure 2 jcm-09-02338-f002:**
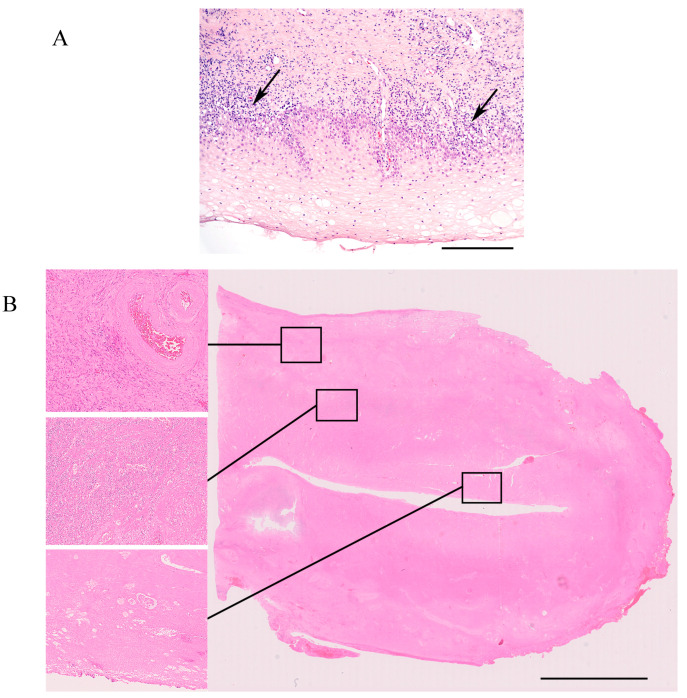
(**A**) Ectocervical biopsy 3 months after UTx (patient #1). There is a dense, mixed inflammatory infiltrate at the stromal–epithelial interface. In some areas, the vacuolization of basal epithelial cells is seen (arrows). These morphological changes are consistent with Grade 1 rejection. Bar = 250 micrometer. (**B**) Uterine graft explanted 1 month after UTx (patient #8). The large picture shows a scanned whole mount slide of the uterine cavity and wall. The left-hand side panel of higher magnification pictures shows (top) the outer myometrial wall with viable myometrium and an artery containing red blood cells. Middle: myometrium with dense neutrophilic infiltrate and necrotic tissue. Bottom: the uterine cavity with necrotic tissue, without viable endometrial glands. Bar = 10 mmeter.

**Table 1 jcm-09-02338-t001:** Characteristics of recipients (R) and matching donors (D). Age is given as completed full years at transplantation. BMI expressed as kg/m^2^.

Recipient	Age/BMI	Recipient Vagina	Donor (Age)	Parity (Birth Weight (g))	CMV D/R	EBV D/R	Blood-Group D/R	HLA mm I/II
#1	22/25.1	dilated	Mother (49)	II (2500, 2700)	+/+	+/+	A/A	0/1
#2	32/20.6	skin	Mother (62)	III (4600, 3680, 4190)	+/+	+/+	O/O	2/1
#3	33/27.6	dilated	Mother (55)	I (2700)	+/+	+/+	A/A	1/0
#4	29/18.0	dilated	Friend (48)	III (3970, 3600, 4355)	−/+	+/+	O/A	3/1
#5	24/24.1	dilated	Mother (45)	III (3100, 3680, 3805)	+/−	+/+	A/A	2/1
#6	30/25.8	dilated	Mother (57)	II (3700, 4100)	+/+	+/+	O/O	2/0
#7	31/20.7	dilated	Sister (37)	III (3500, 3200, 3500)	−/−	+/+	O/AB	2/1
#8	23/26.3	dilated	Mother (46)	IV (3900, 3500, 3500, 3500)	+/−	+/+	A/A	2/0

BMI = body mass index, CMV = cytomegalovirus, EBV = Epstein-Barr virus, HLA mm = human leukocyte antigen mismatch.

**Table 2 jcm-09-02338-t002:** Surgical data of uterus transplantation. EBL = estimated blood loss; LOS = length of hospital stay.

Patient	Total Surgical Time (h)	Reperfusion Blood Flow Left/Right (mL/min)	Total Ischemic Time (h)	No. Vein/Artery Anastomoses	EBL (mL)	LOS (Days)
#1	6	80/40	2.0	3/2	600	5
#2	5.3	50/40	2.0	3/2	400	9
#3	6.6	20/10	3.2	4/2	150	6
#4	4.5	40/50	2.3	3/2	100	6
#5	4.9	35/35	1.7	4/2	100	5
#6	6.0	125/98	1.3	4/2	250	5
#7	5.0	47/31	2.2	4/2	150	5
#8	5.0	39/10	2.0	3/2	100	5

**Table 3 jcm-09-02338-t003:** Post-transplantation monitoring outcomes at 6 months after uterus transplantation.

Patient	Rejection Episodes	Outcome	Hemoglobin Mean (SD)	Creatinine Mean (SD)	Tacrolimus Mean (SD)
#1	1 (grade 1)	RM	126 (7.5)	69 (8.6)	13.3 (2.0)
#2	0	RM	120 (5.6)	54 (4.0)	9.0 (1.9)
#3	–	Hyst	105 (7.8)	72 (12.6)	13.4 (6.5)
#4	1 (grade 2)	RM	117 (7.1)	67 (4.9)	14.2 (4.4)
#5	0	RM	117 (9.0)	60 (5.9)	11.5 (2.7)
#6	0	RM	105 (5.6)	57 (4.9)	11.8 (2.2)
#7	0	RM	117 (6.7)	57 (5.6)	12.2 (3.9)
#8	–	Hyst	105 (12.1)	92 (7.6)	12.3 (3.2)

RM = regular menstruation; Hyst = hysterectomy; – = could not be determined because of ischemic changes; SD = standard deviation.

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
