# Peer review of "Outcome of Recipient Surgery and 6-Month Follow-Up of the Swedish Live Donor Robotic Uterus Transplantation Trial"

_jcm, 2020, doi:10.3390/jcm9082338_

Round 1

Reviewer 1 Report

The authors describe the outcome of recipients after UTx with robotic donor surgery. This study is the largest trial in UTx with robotic donor surgery. Thefore, this manuscript is an extremely meaningful and valuable article for further development of UTx research fields. Moreover, these outcomes include negative data (i.e. graft failure afte UTx ), which is likey to be most important  scientific data because UTx is at experimental stage yet. Thus, this manuscript surely contributes to future knowledge and development in this field.

Please revise the followng minor points to format this article.

  1. "Procure" or "recovery" would be preffred instead of "harvest" in the document.
  2. The section of "Post transplantation follow-up and surgery of graft failure" in M&M does not include concerning surgery of graft failure. Please delete the term of "surgery of graft failure" or give any information of surgery of graft failure.
  3. Please add annotations of  abbreviations of EBL and LOS in Table 2, even these are described in the main text.
  4. Line 310; Please revise the typo, "yhere" to "there".

Author Response

Rev 1

The authors describe the outcome of recipients after UTx with robotic donor surgery. This study is the largest trial in UTx with robotic donor surgery. Thefore, this manuscript is an extremely meaningful and valuable article for further development of UTx research fields. Moreover, these outcomes include negative data (i.e. graft failure afte UTx ), which is likey to be most important  scientific data because UTx is at experimental stage yet. Thus, this manuscript surely contributes to future knowledge and development in this field.

Thanks for positive general comments.

Please revise the followng minor points to format this article.

  1. "Procure" or "recovery" would be preffred instead of "harvest" in the document.

This has been changed throughout the MS

  1. The section of "Post transplantation follow-up and surgery of graft failure" in M&M does not include concerning surgery of graft failure. Please delete the term of "surgery of graft failure" or give any information of surgery of graft failure.

The section heading has now been changed to only ”Post transplantation follow-up”

  1. Please add annotations of  abbreviations of EBL and LOS in Table 2, even these are described in the main text.

These have now been incorporated in the text.

  1. Line 310; Please revise the typo, "yhere" to "there".

Typo is corrected.

Reviewer 2 Report

In this article Brännström and coworkers report on eight cases of uterus transplantation focusing on outcomes at 6 months and factors predictive of early graft failure. This study is provided by a leading group in this emerging field. Reporting on just eight cases (that are still a good number for uterus transplantation) does not allow for sound conclusion, as just feasibility can be shown. As acknowledged by the authors themselves, their techniques are still evolving, which is not surprising.

I have several issues, that I would like to see addressed:

The authors describe that at the back-table they flushed uterine arteries with 10 mL of physiological saline containing heparin (1250 IE). Was sodium heparin given also to the donor before vascular crossclamping?

The authors mentioned that they used HTK for uterus perfusion. What is the background for this choice?

The authors reported two cases of vaginal stenosis and described a method of continuous suture for the vaginal-vaginal anastomosis. While preserving blood flow to vaginal stumps (i.e. to avoid to burn them) is important, the use of continuous sutures, as compared to interrupted sutures, is known to be associated with high rates or strictures in basically all types of anastomosis. Is there a specific reason why not to use interrupted sutures?

Still concerning vaginal stricutures, is sharp transection of the vagina (without the use of any energy device) with hemostasis achieved at the time or reperfusion using absorbable sutures an option?

The authors reported that they injected lidocaine in the uterine arteries at the back-table. Is there some experimental evidence behind this?

The authors reported that all vascular anastomoses were completed before revascularization. Considering that at the back-table you can define if there is a god collateral circulation in the womb, is releasing of vascular clamps after completion of vascular anastomoses on one side an option?

Concerning graft assessment at the back-table, do you see any space for machine-perfusion? How often you have discarded a graft at the back-table and why?

Was donor number 3 a smoker or had history of smoking?

With all the limits of this topic and the limited follow-up, please, add to the discussion about de novo DSA, if investigated, and the immunological follow-up for both the patients in whom the uterus was removed and for the remaining recipients Did you see any differences (from the point of view of DSA), also from the previous experience, in patients with rejection? This could be helpful for a better follow-up.

Even if the article focuses on the recipient, could you please report if any complication occurred in the donor? Did you notice any difference with the previous trial? (Brännström M, Johannesson L, Dahm-Kähler P, et al. The first clinical uterus transplantation trial: a six 391 months report. Fertil Steril 2014; 101:1228-1236.)

Author Response

Rev 2

The authors describe that at the back-table they flushed uterine arteries with 10 mL of physiological saline containing heparin (1250 IE). Was sodium heparin given also to the donor before vascular crossclamping?

No, heparin was not given to donor before cross-clamping.

We have chosen not to include any discussion concerning this in the revised MS.

The authors mentioned that they used HTK for uterus perfusion. What is the background for this choice?

HTK was used in most of our studies with large animals (sheep, baboon) and we have continued to use it in the human setting.

We have chosen not to include any discussion concerning this in the revised MS but simply stating what has been used.

The authors reported two cases of vaginal stenosis and described a method of continuous suture for the vaginal-vaginal anastomosis. While preserving blood flow to vaginal stumps (i.e. to avoid to burn them) is important, the use of continuous sutures, as compared to interrupted sutures, is known to be associated with high rates or strictures in basically all types of anastomosis. Is there a specific reason why not to use interrupted sutures?

Interrupted sutures could be used, but since we used continuous suture (according to our routine for vaginal vault closure in standard hysterectomy) in the initial study of 2013, with no stenosis problems occurring then, we continued to use this suture technique.

We have chosen not to include any discussion concerning this in the revised MS.

Still concerning vaginal stricutures, is sharp transection of the vagina (without the use of any energy device) with hemostasis achieved at the time or reperfusion using absorbable sutures an option?

That would be an option, but it may be difficult to achieve hemostasis on the back side of the vagina, since this part is inaccessible after reperfusion, due to that the graft should not be moved too much in order to not compromise blood flow to the organ.

We have chosen not to include any discussion concerning this in the revised MS.

The authors reported that they injected lidocaine in the uterine arteries at the back-table. Is there some experimental evidence behind this?

There is no experimental evidence for this but local tradition in our kidney transplantation procedures, based on that lidocaine in high doses can have a vasodilation effect.

We have chosen not to include any discussion concerning this in the revised MS.

The authors reported that all vascular anastomoses were completed before revascularization. Considering that at the back-table you can define if there is a god collateral circulation in the womb, is releasing of vascular clamps after completion of vascular anastomoses on one side an option?

Yes this would be an option. When needed for correction of anastomoses on one side the other side can remain open with good flow to the whole uterus as we have experienced several times. However the time of completion of the second side is around 20-30 min and the graft can remain cool during this time. We don’t think this time frame would add any significant ischemic injury and prefer to not have any bleeding from the vaginal cuff or surrounding tissue that may interfere with the visualisation during completion of anastomosis.

We have chosen not to include any discussion concerning this in the revised MS.

e

Concerning graft assessment at the back-table, do you see any space for machine-perfusion?

Machine perfusion may provide information on the capacity of the vascular perfusion and also assist in reducing the peripheral resistance. Considering the NNT in achieving evidence of a better clinical outcome that has been reported in other organs it may take some time to prove this for UTx but we support the idea.

We have chosen not to include any discussion concerning this in the revised MS.

How often you have discarded a graft at the back-table and why?

We have not done this in Sweden, but been involved in one case in Tübingen, Germany (Ref 15).

Was donor number 3 a smoker or had history of smoking?

She was a never-smoker. We have added information about smoking habits in the MS (first paragraph of Results). There was only one previous smoker (# 5 – occasional smoker). Thus, the donors of the grafts failures were both never-smokers.

With all the limits of this topic and the limited follow-up, please, add to the discussion about de novo DSA, if investigated, and the immunological follow-up for both the patients in whom the uterus was removed and for the remaining recipients Did you see any differences (from the point of view of DSA), also from the previous experience, in patients with rejection? This could be helpful for a better follow-up.

 We have not yet analysed this in detail. The first case of graft loss in this series has been evaluated for re transplantation and has DSA and is positive in 50% in a PRA. This analysis was done after the 6 months interval covered in the present study. We have so far not had any clinical use for the finding of DSA and we have not had any indications of humoral rejection.

The findings of DSA development from our original study (2013) and the present study will be published in later publications with long-term follow-up results.

We have commented on this issue together with a comment on the rejection pattern in the Discussion (paragraph 4), as well as that information on DSA analyses and results of negative DSA at UTx, are included in MM and Results.

Even if the article focuses on the recipient, could you please report if any complication occurred in the donor? Did you notice any difference with the previous trial? (Brännström M, Johannesson L, Dahm-Kähler P, et al. The first clinical uterus transplantation trial: a six 391 months report. Fertil Steril 2014; 101:1228-1236.)

The two donor complications (reversible pressure alopecia and pyelopnephritis) are reported in our study of donor surgery of this UTx cohort (ref 11. Brännström M, Kvarnström N, Groth K, et al. Evolution of surgical steps in robotic-assisted donor surgery for uterus transplantation: results of the eight cases of the Swedish study. Fertil Steril 2020; in press).